# Plancraft: an evaluation dataset for planning with LLM agents

**Gautier Dagan & Frank Keller & Alex Lascarides**
University of Edinburgh
{gautier.dagan, keller, alex}@ed.ac.uk

## Abstract

We present Plancraft, a multi-modal evaluation dataset for LLM agents. Plancraft has both a text-only and a multi-modal interface, based on the Minecraft crafting GUI. We include the Minecraft Wiki to evaluate tool use and Retrieval Augmented Generation (RAG), as well as a handcrafted planner and Oracle Retriever, to ablate the different components of a modern agent architecture. To evaluate decision-making, Plancraft also includes a subset of examples that are intentionally unsolvable, providing a realistic challenge that requires the agent not only to complete tasks but also to decide whether they are solvable at all. We benchmark both open-source and closed-source LLMs and compare their performance and efficiency to a handcrafted planner. Overall, we find that LLMs and VLMs struggle with the planning problems that Plancraft introduces, and offer suggestions on how to improve their capabilities.

## 1 Introduction

With the increased performance and affordability of Large Language Models (LLMs) (OpenAI, 2024; Dubey et al., 2024), LLM-based agents have exploded in popularity (Xi et al., 2023). We define LLM agents as systems that incorporate at least one call to an LLM which influences its autonomous decisions about actions in an environment. LLMs are promising for agent-based systems because they can leverage general knowledge acquired during pretraining to tackle tasks expressed in natural language. They can also query and interpret knowledge bases, ask questions, and incorporate new information via text or images, all through a flexible dialogue interface. This potential for human interaction during planning and execution is challenging for other planning paradigms such as Reinforcement Learning (RL) or symbolic planners (Xi et al., 2023). Various methods and strategies (Yao et al., 2023a;b; Shinn et al., 2023) aim to maximise the performance of LLM-based autonomous agents (Xi et al., 2024). But LLMs still exhibit problematic issues that limit their reliability and usefulness for planning—hallucinations and brittleness to inputs, limited context windows, and lack of grounding when placed in new environments.

Most existing benchmarks for evaluating LLM agents in interactive environments focus on success rates (Shridhar et al., 2021; Yao et al., 2022; Liu et al., 2023; Xi et al., 2024): that is, the proportion of trials in which the agent achieves a goal state. Success rates measure whether the agent constructed a valid plan, but do not measure efficiency or the plan's quality. Some environments allow an arbitrary number of incorrect steps, while others immediately terminate an episode if a wrong action is executed. Reporting or measuring only success rates also introduces a dataset bias, as it implies that the solution to a given problem is easily verifiable, yet difficult to obtain. As a result, we argue that **success rate alone is insufficient to capture the complexity of real-world scenarios**. Each additional step of inference incurs non-negligible costs, so metrics should include more fine-grained assessments, such as how close the LLM's plan is to a handcrafted solution. Furthermore, **effective agent-based systems should recognise when a task is unsolvable**, as many real-world tasks may lie beyond the agent's capabilities and indeed many real-world tasks might be difficult to verify as doable, or not. Without the capacity to predict if there's no valid plan, an agent will incur significant costs in continually monitoring and replanning.

To this end, we introduce Plancraft, a new multi-modal planning evaluation dataset based on Minecraft, that constrains the environment to the crafting GUI (see Figure 1). Plancraft consists of planning problems of diverse complexity and length and includes a portion of the dataset that is intentionally unsolvable. Plancraft, unlike previous environments, allows us to benchmark agents against a planner,

| Dataset | Type | Visual Inputs | Knowledge Base | Planner | Impossible Set |
|---|---|---|---|---|---|
| Mind2Web (Deng et al., 2023) | Web | ✗ | ✗ | ✗ | ✗ |
| WebShop (Yao et al., 2022) | Web | ✓ | ✗ | ✗ | ✗ |
| MiniWoB++ (Liu et al., 2018) | Web | ✓ | ✗ | ✗ | ✗ |
| WebArena (Zhou et al., 2023) | Web | ✗ | ✓ | ✗ | ✓ |
| GAIA (Mialon et al., 2023) | Web | ✓ | ✓[†] | ✗ | ✗ |
| TravelPlanner (Xie et al., 2024) | Web | ✗ | ✗ | ✓ | ✗ |
| VirtualHome (Puig et al., 2018) | Home | ✓ | ✗ | ✗ | ✗ |
| ALFWorld (Shridhar et al., 2021) | Home | ✗ | ✗ | ✗ | ✗ |
| ALFRED (Shridhar et al., 2020) | Home | ✓ | ✗ | ✓ | ✗ |
| MineDojo (Fan et al., 2022) | Game | ✓ | ✓ | ✗ | ✗ |
| ScienceWorld (Wang et al., 2022a) | Game | ✗ | ✗ | ✗ | ✗ |
| BabyAI (Chevalier-Boisvert et al., 2019) | Game | ✗ | ✗ | ✓ | ✗ |
| TextCraft (Prasad et al., 2023) | Game | ✗ | ✗ | ✗ | ✗ |
| Plancraft (ours) | Game | ✓ | ✓ | ✓ | ✓ |

Table 1: A comparison of different interactive datasets commonly used in evaluating LLM agents. Each dataset expresses goals or tasks in natural language and requires multi-step planning. We design Plancraft to provide a testing ground for which there exists both a natural language knowledge base and a handcrafted planner. Plancraft also includes data that is intentionally unsatisfiable (the Impossible Set), since the capacity to predict that there's no valid plan is important. [†]GAIA allows tool use, including web search, however a knowledge base on how to solve the task is not provided.

but in a setting that was designed by humans for humans. Since the crafting component of Minecraft is inherently designed for human players, Plancraft also offers a way to test how LLM agents can leverage human knowledge (in the form of the Minecraft Wiki) to solve planning tasks. We compare Plancraft with popular interactive datasets in Table 1.

We evaluate different LLM-based agents on our dataset, thereby providing LLM-agent baselines. We test both open-source and closed-source models and evaluate them against different sets of possible actions. For open-source models, we evaluate the impact of fine-tuning agents on a set of expert plans, and compare multi-modal vs. text-only agents. We also fine-tune a bounding-box detection model on our environment to provide an interface through which text-only LLMs can interact with the multi-modal environment. Finally, we release all baseline models and code along with the dataset and environment as a stand-alone Python package.[1]

## 2 Related Work

### 2.1 LLM Agents

Various strategies have been proposed to leverage LLMs as agents with interactive feedback (Huang et al., 2022; Shinn et al., 2023; Yao et al., 2023a;b; Wang et al., 2022b). The basic idea is to harvest LLMs for information, which, if novel to the agent, may result in better behaviours than they would perform otherwise. The simplest strategy, ReAct (Yao et al., 2023b), consists of interleaving actions with 'thinking' steps where the LLM is allowed unconstrained generation. Other broader strategies, such as Reflexion (Shinn et al., 2023) or Inner Monologue (Huang et al., 2022) try to promote self-corrective behaviour through external modules or steps. Self-Consistency (Wang et al., 2022b) and Tree-of-Thought (Yao et al., 2023a) sample the space of possible paths and select a solution through a majority vote or through another LLM evaluation step. Although all of these techniques are simple to implement, they often add significant overhead in inference compute.

One limitation of LLMs is the size of their context window. This is often addressed with Retrieval-Augmented Generation (RAG) (Lewis et al., 2020), which restricts the context to only relevant examples (by some quantitative metric) from the larger dataset, given the current observations and task. For example, contexts in Voyager (Wang et al., 2023a) consist of the most likely applicable skills for a given problem; those in JARVIS-1 (Wang et al., 2023b) are successful trajectories that are sufficiently similar to current observations. Since crafting is a core part of Minecraft, its Wiki contains a page with details of recipes for crafting all items. The Minecraft Wiki is therefore well suited to a RAG pipeline where an agent can query the Wiki as a knowledge source.

---

[1]https://github.com/gautierdag/plancraft

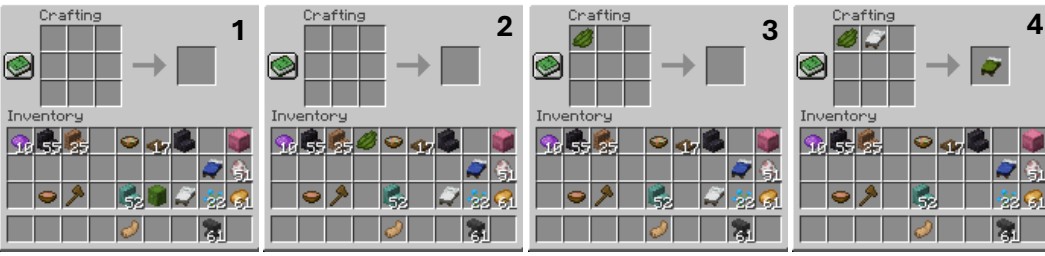

**Task:** Craft a green bed [ 🛏️ ]

**1. Smelt** cactus [ 🌵 ] into green dye [ 🟩 ]    **3. Move** white bed [ 🛏️ ] into grid

**2. Move** green dye [ 🟩 ] into grid    **4. Move** green bed [ 🛏️ ] into inventory

Figure 1: A Plancraft example where the task is to craft a green bed 🛏️. The agent has to use the observations (text or image) to generate the next action. In this case, the shortest path is: crafting the green dye 🟩 by smelting the cactus 🌵; then moving the items (white bed 🛏️ and green dye 🟩) into the correct crafting slots; then moving the resulting green bed 🛏️ into the inventory.

Enabling LLMs to use external tools, such as web search, calculators, and database queries, can also significantly expand their capabilities. This has inspired numerous research efforts (Nakano et al., 2022; Schick et al., 2023; Parisi et al., 2022; Yang et al., 2023; Patil et al., 2023) to integrate and evaluate tool use (or external actions) into LLM agents. Some datasets, such as GAIA (Mialon et al., 2023), even evaluate agents without any restriction on the tools allowed.

Beyond RAG and tool use, some work has examined the limitations of LLMs in planning tasks. Kambhampati et al. (2024) argue that while LLMs struggle with long-term planning, they can still complement traditional methods in an LLM-modulo framework. Several datasets like PlanBench (Valmeekam et al., 2023), TRAC (He et al., 2023), and ACPBench (Kokel et al., 2025) were proposed to evaluate planning-related reasoning. ACPBench and TRAC primarily test planning capabilities through multiple-choice or boolean questions, while PlanBench uses LLMs to generate full plans directly given different testing conditions. However, since these datasets do not involve interactive execution, we do not include them in Table 1.

## 2.2 Interactive LLM Evaluation Datasets

Research on LLM agents requires datasets and benchmarks for evaluation. These datasets vary significantly in the types of environments in which actions are executed, ranging from simplified text-only worlds to multi-modal environments. ALFWorld (Shridhar et al., 2021) and ALFRED (Shridhar et al., 2020) provide virtual home environments in which the agent must manipulate objects to fulfil the instruction, expressed in natural language. Mind2Web (Deng et al., 2023), WebShop (Yao et al., 2022), and WebArena (Zhou et al., 2023) evaluate agents in web-based environments, with tasks requiring interaction with browser interfaces or e-commerce platforms. BabyAI (Chevalier-Boisvert et al., 2019) is a 2D maze-solving game with natural language instructions. Minecraft is also a popular benchmark for agent evaluation, with various datasets such as MineRL (Guss et al., 2019), MineDojo (Fan et al., 2022), and TextCraft (Prasad et al., 2023). In Table 1, we compare Plancraft against these LLM evaluation datasets. While these vary in domain and implementation, they all require agents to do multi-step planning in interactive environments to achieve tasks expressed in natural language.

**Knowledge Base.** Minecraft has the advantage that it is a game humans play, rather than constructed only for agent evaluation. So LLM agents can take advantage of existing online content that assists human players. Baker et al. (2022) collected a large dataset of Minecraft videos and trained a model, Video Pretraining (VPT), to solve tasks directly from pixels in MineRL (Guss et al., 2019). Fan et al. (2022) introduced MineDojo along with a knowledge base of scraped resources such as Minecraft videos and pages taken from the Minecraft Wiki and Reddit forum; however, they do not evaluate whether these improve performance. In Plancraft, we collect the pages of recipes on the Minecraft Wiki and use them to build a knowledge base to evaluate RAG capabilities. We also implement an Oracle Retriever for recipes to provide an upper estimate of a well-performing RAG system.

**Planner.** All of the datasets listed in Table 1 can evaluate whether an agent achieves the goal. But evaluating a plan's quality requires a designated optimal policy for comparison. Of the datasets that we compare to, only BabyAI (Chevalier-Boisvert et al., 2019) and ALFRED (Shridhar et al., 2020) release handcrafted policies or solvers. BabyAI implements a Bot Agent with handcrafted policies that can act as a teacher agent for learners in the environment. ALFRED releases expert demonstrations found using a PDDL solver and reports Path Weighted Metrics that take the length of the expert into account. Similarly, we implement a planner for Plancraft to provide a benchmark against which to compare agents (Section 3.4). While PDDL-based classical planning approaches (McDermott et al., 1998) could theoretically be applied to Plancraft, the domain presents unique challenges due to large search space (see Appendix D.1).

**Impossible Set.** Most interactive datasets assume that every task has a solution. So they cannot evaluate mechanisms that assess feasibility or handle inherently unsolvable tasks. This can be particularly costly for strategies like Tree-of-Thought, which expand the search space by extensively sampling the LLM for potential solutions. Plancraft addresses this gap by including a set of intentionally impossible tasks – tasks with no valid plan. This is similar to the task of identifying non-reachable atoms, as proposed in ACPBench (Kokel et al., 2025), or the unachievable tasks in WebArena (Zhou et al., 2023). By incorporating impossible tasks, Plancraft evaluates an agent's capacity to assess task feasibility. This promotes more realistic and cost-effective decision making, encouraging agents to balance problem solving with estimates of feasibility.

## 3 Plancraft

To create Plancraft, we implement the logic and visual representation of the Minecraft crafting GUI. Like Prasad et al. (2023), we reconstruct the crafting process entirely in Python using game files and images from the Wiki: we use image manipulation to overlay items on top of the inventory background. This improves performance compared with using the Java game engine. Our environment has a one-to-one pixel mapping with the real Minecraft interface and supports multiple resolutions.

### 3.1 Assumptions

Plancraft makes a number of simplifying assumptions. First, there is a **single agent** in the environment, and all changes to the environment are a result of this agent's actions. Second, the effects of executable actions are **deterministic**. Third, the crafting interface and inventory are **fully observable**, either through the text description or through the multi-modal image input. And finally, aligned with classical planning, actions are **sequential**: they are discrete and executed one at a time.

### 3.2 Action and Observation Space

The abstraction level chosen for the action and observation space impacts the tractability of the planning problem. Other Minecraft environments such as MineDojo (Fan et al., 2022) and Textcraft (Prasad et al., 2023) provide a high-level 'craft' command that doesn't require the agent to manipulate the underlying ingredients. In contrast, Plancraft requires the correct placement of ingredients in the inventory and, as such, provides a lower-level symbolic action space. The two possible *environment* actions are `smelt` and `move`, and both actions require a `slot_from`, a `slot_to` and a `quantity`. This abstracts control dynamics and focuses on planning and spatial reasoning.

In the Minecraft crafting GUI, there are 46 possible item slots or positions. The *crafting slot* can be populated with an item only if the correct items have been deposited within the $3 \times 3$ *crafting grid*—items can be moved in and out of this grid from the remaining 36 slots, in which items are stored. Items can only be withdrawn from the crafting slot, and doing this confirms the crafting operation and removes the items present in the crafting grid. Crafting recipes in Minecraft are either Shapeless or Shaped. Shapeless recipes only check whether the items required are present in the crafting grid, whereas Shaped recipes require the items to also be placed in the correct positions. When the items on the crafting grid match one of the 634 recipes, then the resulting item is added to the crafting slot. We denote the crafting slot as `[0]`, the $3 \times 3$ crafting grid as slots `[A1]` to `[C3]` where the letter denotes each row and the number denotes the column, and the remaining 36 inventory slots as slots `[I1]` to `[I36]`. See Figure 2 for an example of a text description of the inventory that uses our annotation scheme.

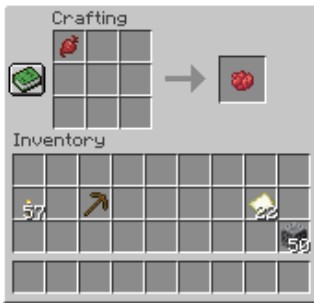

```
inventory:
 - red_dye [0] quantity 1
 - beetroot [A1] quantity 1
 - torch [I10] quantity 57
 - wooden_pickaxe [I12] quantity 1
 - map [I17] quantity 22
 - lodestone [I27] quantity 50
```

Figure 2: Example of a Plancraft observation of the scene in either an image or text format. For the text, we encode the scene using a specific slot notation where [0] denotes the crafting slot, [A1] to [C3] denotes each row and column of the 3 × 3 crafting grid, and [I1] to [I36] denotes the inventory slots.

Plancraft provides two types of observations: text-only and multi-modal. In **text-only**, agents receive a text description of the current inventory and crafting goal and the inventory is perfectly represented using our annotation scheme. Whereas in **multi-modal**, agents receive an image (dimensions 664 × 704) that perfectly contains the crafting GUI, while the goal is provided as text.

### 3.3  Task Setup

The tasks within Plancraft all involve crafting specific items using a predefined set of available resources. The goal of each task is expressed in natural language, such as '*Craft an item of type: green_bed*'. The complexity of these tasks varies, ranging from single-step crafts (e.g., crafting wooden planks 🟫) to multi-step plans that require chaining multiple crafting recipes and actions (e.g., crafting a bed 🛏 first requires planks 🟫 and wool ⬜).

To generate crafting tasks, we represent the dependencies among items as trees. The root of a tree is the target item while its leaves and intermediate nodes are the materials required to craft that item. For each crafting task, we sample the required recipes for the target item and then recursively explore the necessary materials to craft it. This builds a step-by-step plan from raw materials to the final product, simulating the crafting processes with varying levels of complexity. In addition to the necessary crafting materials, we also sample a number (4, 8, 16) of distractor items to make identifying the correct crafting path more challenging. Finally, we validate that all generated tasks can be solved in the text-only and multi-modal environments using a planner (see Section 3.4). We exclude invalid examples and ones where the handcrafted planner's trajectory exceeds 30 steps.

#### 3.3.1  Dataset Statistics

We define a complexity metric as directly proportional to the number of items used and the number of recipes needed to craft the item. We sample trajectories from least to most complex (easy, medium, hard), where hard examples can always be completed in less than 30 steps (see Appendix A.2 for details). Our complexity correlates with the planner's search space exploration, with more complex tasks requiring more steps. We stop episodes after a maximum number of steps $n = 30$ is reached.

**Impossible Tasks.** Since we also wish to evaluate whether agents can predict when a task is unsolvable, we include an additional portion (17%) of the dataset that requires the agent to craft an item, which is impossible given the inventory. To generate this dataset, during sampling, key materials from the path are deliberately removed from the inventory. This is designed to test an agent's ability to recognise when a task cannot be completed given the available resources. As mentioned, all previous interactive environments (see Table 1) rely on the assumption that the task given to an agent is solvable. However, real-world scenarios are unlikely to meet this assumption and tasks may often be unsolvable due to missing resources or constraints.

In total, we sample 1145 training examples, 570 validation examples, and 580 test examples. We hold out items such that 79 items in the validation set are not seen in the training set, and similarly 128 items in the test set do not appear in the training set, 63 of which also do not appear in the validation

set. Since these evaluation sets are extensive, we also provide smaller subsets of the validation (110 examples) and test set (117 examples where 20 are impossible) which we use in our evaluation.

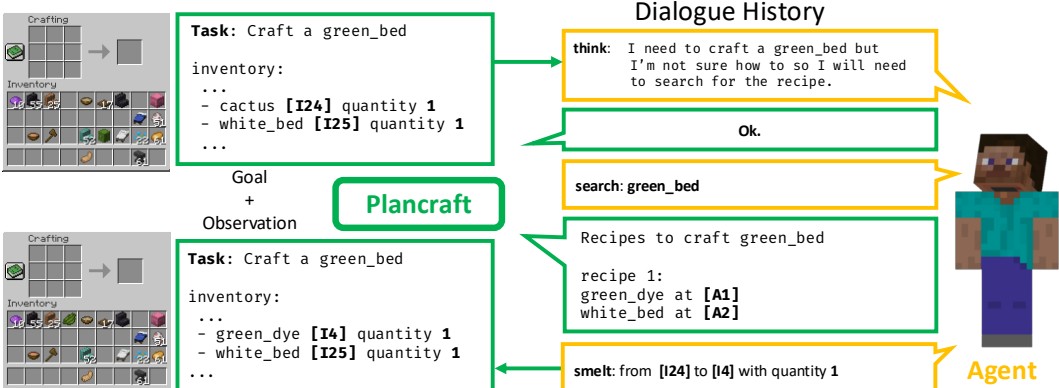

Figure 3: Example flow of Plancraft. The agent can use the varying set of tools. The dialogue history is passed as a sequence of messages. Environment messages include the task goal and current observation (text or image). If the agent takes an action that does not impact the environment such as `think` or `search`, then the environment does not return an observation but a response based on this action.

### 3.3.2 Environment Feedback

Since all models in Plancraft can handle text, we propagate formatting errors as text observations. This feedback informs agents about specific issues, allowing agents to adjust accordingly. For example, when an agent tries to move an item between identical source and destination slots, the environment returns a message specifying that different slots must be used. Or, if an agent generates a quantity that exceeds defined limits, the system provides an error message that the quantity must fall within acceptable parameters. This feedback mechanism allows agents to correct their actions and ensures smoother interactions with the environment.

### 3.3.3 Evaluating Impossible Tasks

To evaluate the agent's predictions on when a task is impossible, we introduce the `impossible` action: when emitted by an agent, this stops the episode. An impossible task is considered successful if and only if the agent emits `impossible`. We report the F1 score of emitting `impossible`, since it is now possible for the agent to interrupt a solvable task (see Section 4.1).

## 3.4 Expert Planner

To evaluate the quality of generated plans (and not just their validity, as measured by success rate), we handcraft a planner to find an efficient sequence of actions required to complete each task. This planner serves as a standard against which the performance of various agent models can be benchmarked.

We implement the solver as a memoized Depth-First search over all possible crafting recipes given an inventory. If found, we return the shortest path between the initial inventory and the target object that needs to be crafted as a series of `smelt` and `move` actions. If the solver does not find a shortest path after 30 seconds, we timeout the search. The planner represents the crafting dependency graph as a directed graph where nodes are items and edges represent the crafting relationship between items. This structure allows for efficient planning by focusing only on recipes that are ancestors of the target item in the dependency graph.

Note that, for efficiency, our planner searches over recipes rather than actions. As a result, shorter paths may still exist if an agent can group actions into one. We provide a more detailed discussion of the planner, including pseudocode and its limitations, in Appendix D.

### 3.5 Integrating External Knowledge

Similarly to Fan et al. (2022), Plancraft includes the Minecraft Wiki as a natural language knowledge base, allowing agents to perform RAG to assist in crafting decisions. This addition enables the agents to use external knowledge, testing their ability to retrieve relevant and multi-modal information and apply it to the planning problem. We scrape the pages from the Minecraft Wiki that contain recipes and post-process the data to convert them to Markdown format.

We implement a `search` action that allows an LLM to search for a given item in the knowledge base. Since we wish to evaluate the limits of RAG, we design the `search` operator to return a gold-label instantiation of a recipe required to craft the particular item. The search mechanism uses an exact match to identify the correct recipe for each item, ensuring that the returned information is always accurate. This approach helps estimate the performance of our system under a perfect retriever and parser. Additional details about our RAG implementation are provided in Appendix E.

## 4 Method

On text-only observations, we evaluate the open models Llama 3.1 8B and 3.3 70B (Dubey et al., 2024), Gemma 3 12B and 24B (Team et al., 2025), Qwen3 30B A3B (Yang et al., 2025), and gpt-4o-mini[2] (OpenAI, 2024). All tested models are chat models, and we format the environment observations and feedback as messages from a user role. We show the interactions between the agent and the Plancraft environment in Figure 3. We evaluate each model in the text-only and multi-modal environments, test making available different sets of actions or tools, and measure the impact of zero-shot versus few-shot versus fine-tuning. We implement the ReAct (Yao et al., 2023b) 'thought' strategy as a tool, where the agent can emit `think` as an action within the environment. Any generated text after `think` is ignored. Notably, Qwen3 30B A3B is the only reasoning model in our evaluation. When using Qwen3, we increase the maximum number of generated tokens per step from 256 to 2048 to account for longer reasoning steps and remove ⟨think⟩ traces after the generation.

**Tool use.** We evaluate multiple sets of different tools outside of the core environment actions (`move` and `smelt`), we test whether including `think`, a recipe Oracle Retriever `search` (RAG) and `impossible` has an impact on the planning capabilities of agents. If a tool is active, we incorporate a description of it within the system prompt and provide an example of its use in the few-shot examples (see Appendix A.3).

**Fine-Tuning.** We fine-tune a Llama 3.1 8B model on 1145 planning trajectories from the training set using LoRA (Hu et al., 2021) ($r = 64$, $\alpha = 32$). These trajectories only consist of Observation, Action pairs directly obtained from our handcrafted planner and only include the `smelt` and `move` actions. As a result, we also test whether the LoRA fine-tuned LLM can take advantage of new actions that it has not been explicitly trained on.

**Image Observations.** We also evaluate models on both text and image inputs. Since Llama models are text-only, we train a Faster R-CNN (Li et al., 2021) to extract item types, quantities, and positions from images (Appendix F). This model is trained separately on randomly sampled inventories. It classifies object types and quantities using bounding box features, achieving 99% slot accuracy and 86% accuracy for correct type and quantity. We then map the extracted symbolic representations to the same format as our text-only observations. The bounding box model benchmarks how much planning ability is lost if an observation is generated from a trained but imperfect visual processing system. For gpt-4o-mini, we evaluate using the predictions from the bounding-box model but also test passing image inputs directly (since gpt-4o-mini supports images). We also evaluate multimodal observations on Qwen 2.5 VL 72B (Bai et al., 2025) and the aforementioned Gemma 3 models.

### 4.1 Metrics

For all methods, we report: (a) the task success rate—whether or not the goal item has been crafted; (b) the plan length—the number of *environment* actions the agent has taken (i.e. `smelt` and `move`); and (c) the total number of tokens used (both input and output combined). Task success evaluates the percentage of tasks completed successfully by the agent and excludes the impossible set.

---

[2]gpt-4o-mini-2024-07-18

Using the expert planner, we can compare an agent's trajectories compared to those of the expert. We calculate the average Action Efficiency (AE) as the average difference between the number of actions in an agent's plan $P$ compared to the length of the path of the expert planner $P^e$. Note we only consider successful plans in the calculation of this metric:

$$\text{Action Efficiency} = \frac{\sum_{i=1}^{N_{\text{successful}}} \left( P_i - P_i^e \right)}{N_{\text{successful}}} \tag{1}$$

We measure whether knowledge, internal and external, is used through the number of `think` and `search` calls. When we evaluate the performance of models when the `impossible` is allowed, we also report the F1 score of emitting the action. Since predicting `impossible` immediately stops the episode, the impossible setup introduces a new form of failure mode where a model can emit an `impossible` when faced with a solvable problem.

Lastly, we track the overall compute efficiency using the total number of tokens used as a proxy.[3] Token usage is impacted by tool use since calling external tools and actions both requires the model to generate additional tokens (to call the tool) but also introduces new information as a result of the external call. We note that in Plancraft the observation tends to dominate token usage, therefore tool-use decreases the overall token usage since it leads the model to see fewer observations on average. However, we also note that tools could hide additional compute costs not factored into simple token usage analysis.

Unless specified, all models are evaluated under the same conditions, with a fixed number of trials per task and the same prompts. We use a temperature of $t = 0.6$ and run five generations per model to report an average on each metric.

| Tools | Model | Success Rate (↑) | | | Overall | Avg. Action Count | | | Impossible F1 (↑) | Avg. Plan Length (↓) | AE (↓) | Avg. Tokens Used (↓) |
|---|---|---|---|---|---|---|---|---|---|---|---|---|
| | | Easy | Medium | Hard | | think | search | impossible | | | | |
| M S | Llama 8B | 0.10 | 0.00 | 0.00 | 0.04 | - | - | - | - | 29.06 | 4.38 | 88.9k |
| M S | Llama 70B | 0.38 | 0.12 | 0.00 | 0.18 | - | - | - | - | 25.57 | 2.38 | 76.3k |
| M S | gpt-4o-mini | 0.16 | 0.00 | 0.00 | 0.07 | - | - | - | - | 28.12 | 0.48 | 82.1k |
| M S | Llama 8B FT | 0.63 | 0.48 | 0.11 | 0.40 | - | - | - | - | 20.29 | 0.14 | 56.5k |
| M S | Gemma 12B | 0.16 | 0.03 | 0.00 | 0.07 | - | - | - | - | 28.12 | 0.85 | 79.4k |
| M S | Gemma 27B | 0.24 | 0.08 | 0.01 | 0.12 | - | - | - | - | 27.13 | 1.85 | 67.5k |
| M S | Qwen3 30B A3B | 0.49 | 0.21 | 0.04 | 0.27 | - | - | - | - | 24.29 | 3.66 | 93.6k |
| M S T | Llama 8B | 0.14 | 0.00 | 0.00 | 0.06 | 11.31 | - | - | - | 28.80 | 6.67 | 68.3k |
| M S T | Llama 70B | 0.45 | 0.30 | 0.02 | 0.26 | 11.79 | - | - | - | 24.34 | 3.32 | 58.7k |
| M S T | gpt-4o-mini | 0.21 | 0.05 | 0.01 | 0.10 | 15.43 | - | - | - | 27.65 | 3.63 | 58.4k |
| M S T | Llama 8B FT | 0.61 | 0.48 | 0.10 | 0.39 | 0.07 | - | - | - | 20.59 | 0.26 | 58.9k |
| M S T | Gemma 12B | 0.20 | 0.06 | 0.00 | 0.10 | 17.53 | - | - | - | 27.82 | 2.85 | 57.2k |
| M S T | Gemma 27B | 0.38 | 0.20 | 0.00 | 0.20 | 15.56 | - | - | - | 25.70 | 3.56 | 56.4k |
| M S T | Qwen3 30B A3B | 0.34 | 0.23 | 0.03 | 0.20 | 15.49 | - | - | - | 25.61 | 2.90 | 73.1k |
| M S T I | Llama 70B | 0.45 | 0.29 | 0.03 | 0.26 | 6.98 | - | 0.43 | 0.45 | 16.81 | 2.75 | 40.3k |
| M S T I | Gemma 27B | 0.30 | 0.15 | 0.00 | 0.16 | 2.82 | - | 0.82 | 0.34 | 7.68 | 2.89 | 14.0k |
| M S T I | Qwen3 30B A3B | 0.29 | 0.17 | 0.01 | 0.16 | 3.04 | - | 0.71 | 0.37 | 8.72 | 1.88 | 25.1k |
| M S T SE | Llama 8B | 0.44 | 0.14 | 0.02 | 0.22 | 6.23 | 3.73 | - | - | 25.23 | 4.48 | 63.0k |
| M S T SE | Llama 70B | 0.91 | 0.83 | 0.31 | 0.67 | 4.80 | 2.51 | - | - | 16.99 | 3.95 | 38.0k |
| M S T SE | gpt-4o-mini | 0.46 | 0.13 | 0.02 | 0.23 | 13.24 | 1.93 | - | - | 25.46 | 5.55 | 55.4k |
| M S T SE | Llama 8B FT | 0.62 | 0.54 | 0.10 | 0.41 | 0.03 | 0.08 | - | - | 20.31 | 0.39 | 60.0k |
| M S T SE | Gemma 12B | 0.53 | 0.30 | 0.03 | 0.29 | 13.09 | 1.85 | - | - | 23.75 | 3.36 | 49.8k |
| M S T SE | Gemma 27B | 0.83 | 0.69 | 0.08 | 0.52 | 8.77 | 2.75 | - | - | 19.57 | 4.06 | 43.6k |
| M S T SE | Qwen3 30B A3B | 0.48 | 0.38 | 0.07 | 0.30 | 11.59 | 0.79 | - | - | 23.60 | 2.84 | 72.9k |
| M S T SE I | Llama 8B | 0.28 | 0.03 | 0.00 | 0.12 | 2.06 | 1.81 | 0.85 | 0.32 | 8.29 | 2.90 | 15.4k |
| M S T SE I | Llama 70B | 0.88 | 0.82 | 0.30 | 0.65 | 4.57 | 2.42 | 0.08 | 0.71 | 16.30 | 3.96 | 36.7k |
| M S T SE I | gpt-4o-mini | 0.40 | 0.13 | 0.02 | 0.20 | 6.90 | 1.20 | 0.60 | 0.39 | 15.67 | 5.12 | 32.6k |
| M S T SE I | Llama 8B FT | 0.62 | 0.55 | 0.10 | 0.41 | 0.01 | 0.12 | - | 0.00 | 20.24 | 0.38 | 60.3k |
| M S T SE I | Gemma 12B | 0.43 | 0.28 | 0.02 | 0.25 | 2.91 | 0.58 | 0.71 | 0.36 | 9.05 | 2.87 | 19.1k |
| M S T SE I | Gemma 27B | 0.78 | 0.62 | 0.07 | 0.48 | 2.73 | 1.22 | 0.51 | 0.45 | 8.90 | 3.80 | 17.5k |
| M S T SE I | Qwen3 30B A3B | 0.32 | 0.28 | 0.03 | 0.20 | 1.69 | 0.15 | 0.68 | 0.38 | 7.88 | 2.33 | 25.5k |

Table 2: Average Success Rates (SR) for different models and sets of actions available: `move` (M), `smelt` (S), `think` (T), `search` (SE), `impossible` (I). We also report the average count of special actions used, the average plan length, the Action Efficiency (AE) and the average total number of tokens used.

---

[3] We consider input and output tokens as equivalent.

| Model (M S) | Overall Success Rate (↑) | Avg. Plan Length (↓) | AE (↓) | Avg. Tokens Used (↓) |
|---|---|---|---|---|
| Llama 8B R-CNN | 0.00 | 30.00 | - | 84.5k |
| Llama 70B R-CNN | 0.13 | 27.03 | 2.97 | 79.5k |
| gpt-4o-mini R-CNN | 0.04 | 28.93 | 3.31 | 82.3k |
| Gemma 3 12B IMG | 0.00 | 29.96 | 0.00 | 137.7k |
| Gemma 3 27B IMG | 0.00 | 29.94 | 1.00 | 117.6k |
| Qwen 2.5 VL 72B IMG | 0.01 | 29.89 | 0.50 | 181.1k |
| gpt-4o-mini IMG | 0.02 | 29.45 | 2.50 | 11.6M |

Table 3: Results for different models with the basic sets of actions (M S) and image observations. We use our R-CNN to extract the symbolic representation from images. Since gpt-4o-mini and Gemma 3 also supports image inputs, we evaluate these models as a unified VLM where observations are passed directly as images (IMG). We additionally test the Qwen 2.5 VL 72B model on image only observations.

# 5 Results

We report results with text-only observations in Table 2 and multi-modal observations in Table 3.

**Act and ReAct baselines.** We first restrict the action space to only the environment actions, `move` (M) and `smelt` (S), and evaluate the planning capabilities of LLMs in a few-shot setting. We find that Llama 70B outperforms gpt-4o-mini, with overall task success rates of 0.18 and 0.07 respectively. Llama 8B performs noticeably worse with a success rate of only 0.04. Gemma models perform similarly to gpt-4o-mini with success rates of 0.07 (12B) and 0.12 (27B), while the Qwen3 30B A3B reasoning model achieves 0.27. However comparing Qwen3 is the best model when `think` is not available is not a fair comparaison as Qwen3 is allowed to generate a reasoning trace before answering. In terms of AE, we find that gpt-4o-mini is closer to the expert planner (0.48) than the Llama models (4.38 and 2.38), which indicates that when the model succeeds, it does so efficiently.

We then extend the pool of possible actions to include `think`, similar to a ReAct strategy (Yao et al., 2023b). Note that we also update the few-shot prompt appropriately to include a description of the action and an example of its use (see Appendix A.3). Using `think` improves overall task success in all few-shot models: Llama 8B 0.06 (+0.02), Llama 70B 0.26 (+0.08), gpt-4o-mini 0.10 (+0.03). The Gemma models also show modest improvements, while Qwen3 30B A3B slightly decreases to 0.20 despite its reasoning capabilities. Explicitly including a `think` action in addition to its reasoning trace decreases Qwen3's overall success rate.

**External Knowledge.** When we allow `search`, we enable the models to incorporate the external knowledge contained in the Minecraft Wiki through the RAG Oracle Retriever. We find that this has substantial benefits for all three few-shot models, boosting Llama 8B to 0.22 (+0.16), Llama 70B to 0.67 (+0.41) and gpt-4o-mini to 0.23 (+0.13) in overall success rate. Gemma 27B achieves 0.52 with search, demonstrating significant improvement, while Qwen3 30B A3B reaches 0.30. Instead of speculating on how to craft a particular recipe, `search` allows models to directly retrieve precise information and where items must be placed to complete a recipe. This reduces the need for additional reasoning steps, resulting in more efficient token usage.

**Predicting Impossible Tasks.** Including `impossible` into the action inventory introduces a way for the agent to interrupt an episode. This negatively affects the overall success rate of all models. Having large impact on smaller Llama 8B model (−0.10) and the Qwen3 30B A3B model (−0.30), while having a smaller impact on the larger Llama 70B (−0.02) and gpt-4o-mini (−0.03). We also record the F1 rate of emitting `impossible`. The Llama 70B model outperforms all other models on predicting impossible tasks (F1 score 0.71). Adding `impossible` also decreases token use, since models can now interrupt an episode when stuck. This reduces the average tokens used significantly for some models; for example, Llama 8B's average token use reduces from 63.0k to 15.0k and gpt-4o-mini's from 55.4k to 32.6$k$. We note that predicting `impossible` is not trivial without external knowledge, Llama 70B only gets 0.45 F1 when `search` is not available.

**Effects of Fine-tuning.** As mentioned in Section 4, we also fine-tune an LLM on expert plans (Llama 8B FT). As in Wang et al. (2022a), we find that smaller models can increase their task success with only about 1k training examples. Llama 8B FT obtains an overall success rate of around 0.40, which is remains steady with each additional action we introduce. We find that fine-tuning severely decreases the model's ability to use new actions: Llama 8B FT almost never uses `think` and `search` actions, and never emits `impossible`.

**Image Observations.** In Table 3, we use images instead of text inputs. We use the pre-trained bounding-box Faster R-CNN model to map images into text descriptions, and confirm a drop in accuracy for all models: Llama 8B ($-0.04$), Llama 70B ($-0.05$), and gpt-4o-mini ($-0.03$). Because gpt-4o-mini supports image inputs, we also evaluate passing images, and no text descriptions, directly to the model. This is gpt-4o-mini IMG: the system instructions remain the same, but the few-shot prompt replaces the observations with images. This model cannot solve our task (0.02) and yet uses significantly more tokens, because every image is converted into a long sequence of tokens ($\approx 200$ per $512 \times 512$ image patch). Note that token usage in gpt-4o-mini IMG is an overestimation due to vision tokens being priced at the gpt-4o rate ($10\times$). However, the average token usage remains a direct representation of the average cost per episode.

We also run same image only observations for the Gemma 3 12B/27B and Qwen 2.5 VL 72B class of models, and find similarly that accuracy remains 0 for both Gemma 3 sizes, and Qwen 2.5 VL 72B only manages 1% overall success. To solve Plancraft, a VLM needs to discern where and what objects are in an image, how these change over time, and how to map between these and the text locations/types. VLMs like Gemma 3 26B might approximate positions (e.g., "bottom left") but fail to map items to exact grid locations. As Plancraft can contain dialogue sequences of up to 30 observations, VLMs need to be able to handle 30 different image inputs. We hypothesize that this is in part due to the pre-training and fine tuning regime of VLMs which are rarely trained on long image sequences nor to ground arbitrary game UIs. Overall, our results also demonstrate that prompting VLMs using few-shot prompting in the same way as with text does not work in Plancraft. Modern VLMs are unable to leverage the few-shot image examples to calibrate their predictions and generalise more broadly to new tasks.

## 6 Conclusion

We introduced Plancraft, a multi-modal dataset to evaluate the planning capabilities of LLM-based agents. Using Minecraft crafting, Plancraft enables assessment of both task accuracy and an agent's ability to judge task feasibility. Our results show that larger models, such as Llama 70B, outperform smaller models like Llama 8B in both task success and efficiency in Plancraft. This advantage is amplified by specialised actions such as `think` and `search`. The introduction of `impossible` highlights a trade-off: while enabling models to recognise unsolvable tasks reduces token usage and increases efficiency, it disproportionately impacts smaller models like Llama 8B. Fine-tuning smaller models, as shown with Llama 8B FT, can boost task success but also reveals a key limitation: the inability to effectively use new actions later on like `think` and `search`, indicating fine-tuning may overly constrain agentic behaviour.

Our results highlight challenges in few-shot multi-modal interactive settings. While text-only tasks exhibit strong performance trends, raw image inputs (e.g., gpt-4o-mini IMG) underscore the limitations of out-of-the-box VLMs for effective interactive planning. All models are also sensitive to observations, and even with our pre-trained bounding box model, we observe significant performance degradation as the observations become noisier. Overall, our results underscore the complexity of bridging multi-modal inputs and decision-making in planning. We hope Plancraft aids future work in enhancing tool adaptability and developing RAG systems to handle noisy real-world knowledge.

## Acknowledgments

This work was supported in part by the UKRI Centre for Doctoral Training in Natural Language Processing, funded by the UKRI (grant EP/S022481/1) at the University of Edinburgh, School of Informatics and School of Philosophy, Psychology & Language Sciences and by the UKRI-funded TAS Governance Node (grant number EP/V026607/1).

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

# A  Additional Dataset Details

In this section, we provide additional details about the Plancraft dataset, including the motivation for using Minecraft, the dataset splits, and the complexity metric used to categorize tasks.

## A.1  Why Minecraft?

Since its introduction to AI research through platforms like Malmo (Johnson et al., 2016), Minecraft has evolved into a widely adopted benchmark for evaluating both reinforcement learning algorithms (Guss et al., 2019; Perez-Liebana et al., 2019) and, more recently, language-based agents (Fan et al., 2022; Wang et al., 2023a; Prasad et al., 2023; Wang et al., 2023c). Whereas most previous work deals with the navigation and exploration aspects of Minecraft, abstracting away the crafting steps to simple callable functions, our work focuses entirely on the crafting aspect of Minecraft, which requires a different set of planning and spatial reasoning skills.

Unlike synthetic environments designed specifically for AI research (e.g., ALFWorld (Shridhar et al., 2021) or ALFRED (Shridhar et al., 2020)), Minecraft is a commercially successful game developed for human players. This means that provides an environment with inherent complexity appropriate for human problem-solving capabilities. Additionally, its popularity means that Minecraft comes with extensive human-authored documentation, tutorials, and community knowledge bases that LLMs may have encountered during pretraining (Fan et al., 2022). Minecraft allows us to evaluate how effectively LLMs can leverage existing human knowledge sources (e.g. the Minecraft Wiki) to solve planning tasks, paralleling real-world scenarios where agents must use reference documentation to accomplish goals.

## A.2  Dataset Splits

We split the dataset into training, validation, and test sets, with a total of 1145, 570, and 580 examples respectively. We also provide a smaller validation and test set with 110 and 117 examples respectively, which we use in our evaluation.

Our complexity metric is proportional to the number of items used and the number of recipes required to solve an example (see Figure 4). The complexity score therefore correlates with how many steps are required to solve a task, with more complex tasks requiring more steps. We group examples into five quantiles or complexity bins: very easy, easy, medium, hard, very hard. We also include a set of impossible tasks, which are intentionally unsolvable.

Note that when reporting results, we group together very easy and easy examples together, and the hard and very hard examples together for simplicity. Table 4 shows the distribution of examples across complexity score for each dataset split.

## A.3  Prompts

We provide the system prompt used for all Plancraft experiments in Figure 5. The prompt includes a detailed description of the crafting grid, inventory slots, and available actions. Examples of the format for each action type are included in the system prompt. Figure 6 shows the few-shot (two-shot) interaction examples used throughout the evaluation.

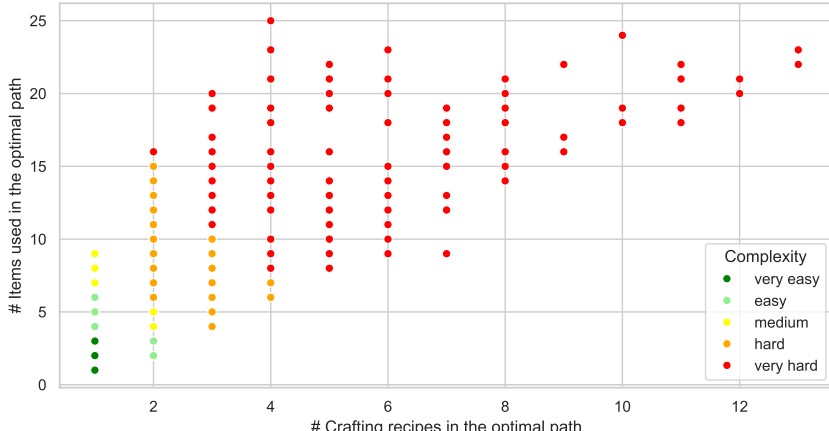

Figure 4: Scatterplot showing the number of items used in a task versus the number of recipes required to craft the target item. The hue of the scatterplot represents the complexity bins.

| Complexity | Train | Val | Test | Val (small) | Test (small) | Val (easy) | Test (easy) |
|---|---|---|---|---|---|---|---|
| very easy | 200 | 100 | 100 | 20 | 20 | 100 | 100 |
| easy | 200 | 100 | 100 | 20 | 20 | - | - |
| medium | 198 | 100 | 100 | 20 | 20 | - | - |
| hard | 200 | 100 | 100 | 20 | 20 | - | - |
| very hard | 147 | 70 | 80 | 10 | 17 | - | - |
| impossible set | 200 | 100 | 100 | 20 | 20 | - | - |
| Total | 1145 | 570 | 580 | 110 | 117 | 100 | 100 |

Table 4: Dataset statistics per complexity score for each split.

## A.4 Environment Feedback

Figure 7 shows the error messages returned by the environment for each action type.

On the other hand, when the agent performs a valid action, the environment responds with a message containing the goal and observation. This means that a valid action that has no effect on the environment will still return a message with the goal and observation, but no additional feedback. Since a no-op action is not penalized, the agent should recognise when it its actions are not having the desired effect and adjust its strategy accordingly. Note, we also do not provide any rewards to the agent. However, it would be possible in the future to include a reward signal based on the number of remaining steps needed to complete the task, or whether the agent is successful.

## B Qualitative Examples

In Figures 8 to 12, we provide qualitative examples of the Llama 8B model interacting with the Plancraft environment. These examples illustrate the model's reasoning process and the impact of different actions on task completion. Figures 8 to 11 are all taken from the same Plancraft task, where the model is asked to craft light_gray_glazed_terracotta. By our complexity categories this is a very easy task as it only requires smelting light_gray_terracotta into a free inventory slot.

In Figure 8, the model has access to `think`, and it successfully completes the task by correctly identifying the required smelting action in its first think step, though we note that it also hallucinates other non-needed actions. Whereas, Figure 9, the model, with the same input, hallucinates non-existent crafting dependencies and attempts to solve for them, eventually reaching the maximum number of steps. In Figure 10, the model uses the `search` action to find the recipe for light_gray_glazed_terracotta, and then uses `think` to identify the correct smelting action. However when we introduce `impossible` in Figure 11, the model uses `impossible` to end the task early,

---

**System Prompt**

```
You are crafting in Minecraft.  You need to decide on the next
action.

Crafting Grid:  The crafting table is organized into a 3x3 grid.
Each slot in the grid has a unique identifier:
- Top row:   [A1] [A2] [A3]
- Middle row:  [B1] [B2] [B3]
- Bottom row:  [C1] [C2] [C3]

The output of the crafting process is placed in a designated output
slot labeled [0].  You cannot move or smelt items directly into
slot [0].

Inventory Slots:  The remaining inventory slots (outside of the
crafting grid) are used for storing items.  These slots are labeled
as [I1] to [I36].

Actions:
     • move:  Transfer a specific quantity of an item from one slot
       to another
     • smelt:  Smelt an item in a furnace and moves the output to a
       specific slot
     • think:  Generate thoughts to help you decide on the next
       action
     • search:  Search for recipes to craft a specific item
     • impossible:  Stop task if it is certain that it is impossible
       with given inventory
Format:
     • 'move:  from [Source] to [Target] with quantity N'
     • 'smelt:  from [Source] to [Target] with quantity N'
     • 'think:  <thought message>'
     • 'search:  <recipe name>'
     • 'impossible:  <reason>'
Example:
     • 'move:  from [I2] to [A1] with quantity 3'
     • 'smelt:  from [I5] to [I6] with quantity 1'
Constraints:
     • You cannot move or smelt items into [0]
     • If an item is not in slot [0] then the recipe is incorrect
     • You need to move items from [0] to a free inventory slot to
       complete the crafting process
```

---

Figure 5: System prompt used for all Plancraft experiments. The coloured actions (think, search, impossible) are included or excluded based on the experimental configuration.

even though it still correctly identified the recipe for light_gray_glazed_terracotta and has the same inventory as in the previous examples. We find models eager to emit `impossible` when it is available, even when the task is solvable, therefore decreasing their overall accuracy.

Finally, in Figure 12, the model is asked to craft diorite_wall, which is an impossible task given the inventory. In this case, the model correctly uses a combination of both `search` and `think` to identify the missing ingredient (diorite) and before outputting `impossible`.

| Tools | Model | Smelting | Shapeless | Shaped | Mixed |
|---|---|---|---|---|---|
| M S | Llama 70B | 0.91 | 0.51 | 0.12 | 0.07 |
| M S T | Llama 70B | 0.69 | 0.57 | 0.25 | 0.12 |
| M S T SE | Llama 70B | 1.00 | 0.80 | 0.80 | 0.42 |
| M S T SE I | Llama 70B | 1.00 | 0.78 | 0.79 | 0.39 |
| M S | Llama 8B | 0.09 | 0.20 | 0.01 | 0.02 |
| M S T | Llama 8B | 0.18 | 0.23 | 0.02 | 0.03 |
| M S T SE | Llama 8B | 0.98 | 0.37 | 0.22 | 0.10 |
| M S T SE I | Llama 8B | 0.66 | 0.24 | 0.10 | 0.06 |
| M S | Llama 8B FT | 0.88 | 0.77 | 0.41 | 0.22 |
| M S T | Llama 8B FT | 0.65 | 0.81 | 0.42 | 0.19 |
| M S T SE | Llama 8B FT | 0.86 | 0.77 | 0.43 | 0.21 |
| M S T SE I | Llama 8B FT | 0.86 | 0.78 | 0.44 | 0.20 |
| M S | gpt-4o-mini | 0.58 | 0.26 | 0.00 | 0.04 |
| M S T | gpt-4o-mini | 0.51 | 0.34 | 0.04 | 0.05 |
| M S T SE | gpt-4o-mini | 0.71 | 0.32 | 0.26 | 0.09 |
| M S T SE I | gpt-4o-mini | 0.76 | 0.26 | 0.24 | 0.07 |

Table 5: Average success for all models and tools when aggregated over the type of recipes needed to craft the target item. The Mixed category requires combining multiple different types of recipes.

| Tools | Model | Avg. # of Invalid Actions |
|---|---|---|
| M S | Llama 70B | 2.88 |
| M S T | Llama 70B | 0.03 |
| M S T SE | Llama 70B | 0.02 |
| M S T SE I | Llama 70B | 0.01 |
| M S | Llama 8B | 6.85 |
| M S T | Llama 8B | 0.20 |
| M S T SE | Llama 8B | 0.44 |
| M S T SE I | Llama 8B | 0.04 |
| M S | Llama 8B FT | 0.02 |
| M S T | Llama 8B FT | 0.00 |
| M S T SE | Llama 8B FT | 0.01 |
| M S T SE I | Llama 8B FT | 0.01 |
| M S | gpt-4o-mini | 1.53 |
| M S T | gpt-4o-mini | 0.02 |
| M S T SE | gpt-4o-mini | 0.04 |
| M S T SE I | gpt-4o-mini | 0.00 |

Table 6: Average Number of Invalid Actions that trigger corrective environment feedback per trajectory for each tool/model configuration.

## C Additional Results

In Table 5, we aggregate success over the different types of recipe categories. We find that Smelting is the easiest type of recipe to craft or solve for since it only requires using the `smelt` action. The hardest type of task is Mixed, when different recipe types are required to craft the target object. We can also observe that Shaped recipes are particularly hard when the `search` action is not available since it requires reasoning over the spatial positioning of items without an relevant example. However with `search`, we find that larger LLMs such as Llama 70B obtain high accuracies since they can more accurately pattern match from the recipe search to the desired crafting positions.

In Table 6, we show the average number of invalid actions that trigger error feedback from the environment. Overall we observe that the non-fine-tuned models do not respond well to the baseline

regime where only `move` and `smelt` are allowed, likely since most have been instruction fine-tuned with some level of chain of thought reasoning. Simply introducing the `think` action greatly helps the each model's ability to follow the formatting of the response.

## C.1 Zero-shot

We compare the zero-shot capability for Llama 70 B on our full tool set (M S T SE I) in Table 7. Overall, we find a significant drop in overall accuracy ($-0.12$) when moving to a zero-shot setting.

| | Model | Success Rate ($\uparrow$) | | | | Avg. Action Count | | | Impossible F1 ($\uparrow$) | Avg. Plan Length ($\downarrow$) | AE ($\downarrow$) | Avg. Tokens Used ($\downarrow$) |
|---|---|---|---|---|---|---|---|---|---|---|---|---|
| | | Difficulty | | | Overall | | | | | | | |
| | | Easy | Medium | Hard | | think | search | impossible | | | | |
| Zero-shot | Llama 70B | 0.75 | 0.57 | 0.26 | 0.53 | 4.36 | 1.63 | 0.17 | 0.58 | 17.63 | 4.98 | 32.6k |
| Few-shot | Llama 70B | 0.88 | 0.82 | 0.30 | 0.65 | 4.57 | 2.42 | 0.08 | 0.71 | 16.30 | 3.96 | 36.7k |

Table 7: Comparing zero-shot versus few-shot prompting on the Llama 70B with the full set of actions available: `move` (M), `smelt` (S), `think` (T), `search` (SE), and `impossible` (I)

# D Planner

In this section, we provide additional details about the Plancraft planner, including the limitations of classical PDDL planners for the Plancraft domain and the implementation details of our custom planner.

## D.1 Plancraft and Classical PDDL Planners

While Plancraft satistifies the requirements of a classical planning domain having deterministic actions and fully observable states, it is not easily representable in PDDL (McDermott et al., 1998). For one, Minecraft's crafting system contains actions in which a parametrised number of items are moved or smelted. While variable amounts are handled in PDDL 2.1+ through the use of numeric fluents, the complexity of representing these actions in PDDL is non-trivial and unsupported by many classical planners such as Fast Downward (Helmert, 2006).

Additionally, the search space of Plancraft is large when compared to traditional planning domains (e.g. Blocksworld), due to the number of possible crafting recipes (859), the number of items (976), possible positions (46), and possible quantities (64). This results in a prohibitively large state space that makes classical planning approaches computationally expensive for all but the simplest Plancraft tasks.

## D.2 Planner Implementation Details

Our planner uses a memoized Depth-First Search (DFS) algorithm to find the shortest path from the initial inventory to the target item. The planner operates in two main phases:

1. Builds a directed graph representing the recipe dependency relationships between items.
2. For a given target item and initial inventory, it performs a search to find the optimal crafting sequence.

The core of the planner recursively explores possible crafting steps while memoizing intermediate results to avoid redundant computation. The pseudocode for this algorithm is presented below:

After finding the shortest sequence of recipes, the planner decomposes each recipe into concrete `move` and `smelt` actions. This decomposition accounts for various recipe types:

- **Smelting Recipes**: Require moving an item to a smelting slot and receiving the output in another slot.

---

**Algorithm 1** Handcrafted Planner Algorithm

---

**Require:** target item, initial inventory, maximum steps, timeout
**Ensure:** shortest sequence of recipes or NULL if impossible
 1: start a timer
 2: identify necessary recipes and sort dependencies
 3: **procedure** DFS(current inventory, steps taken)
 4: **if** timeout exceeded **then**
 5:    **return** NULL
 6: **end if**
 7: **if** target item is in inventory **then**
 8:    **return** steps taken
 9: **end if**
10: **if** steps taken exceed maximum allowed **then**
11:    **return** best known solution
12: **end if**
13: **for** each possible recipe **do**
14:    **if** recipe is feasible **then**
15:      update inventory and steps taken
16:      recursively call DFS with new state
17:      keep track of the shortest valid solution
18:    **end if**
19: **end for**
20: **return** best known solution
21: **end procedure**

---

- **Shapeless Recipes**: Require ingredients to be placed in the crafting grid in any arrangement.
- **Shaped Recipes**: Require ingredients to be placed in specific positions within the crafting grid.

The planner handles inventory management by tracking item quantities and ensuring that crafting operations maintain consistency. This includes removing used ingredients and adding crafted items to the inventory.

### D.3 Limitations of the Planner

While our planner can efficiently solve Plancraft tasks, it does not provide optimal guarantees since it decomposes the search into a series of valid recipes before optimising the action sequence for each recipe. It is possible for shorter plans to exist if the planner were to consider all possible crafting steps simultaneously rather than decomposing the search into individual recipes. For instance, if two recipes share a common ingredient that can be placed in the same slot, the planner may not consider combining the move action for that ingredient across both recipes. The planner also does not `smelt` multiple items in a single action and does not move items directly into the crafting grid from the crafting slot `[0]` or from a `smelt` action These limitations mean that it is possible for an agent in Plancraft to outperform the planner by considering multiple recipes (subgoals) simultaneously. This is why we refer to our planner as a handcrafted planner rather than an optimal planner.

## E  RAG Implementation Details

### E.1  Oracle Retriever

The Oracle Retriever is a simple exact match retriever that returns the recipe for a given item query. Since we have access to the underlying game files which includes all crafting recipes, we can guarantee that the Oracle Retriever always returns a correct recipe for a valid item query. If the item does not exist in the recipe database, the Oracle Retriever returns an error message stating that it could not find a recipe for the item. If there are more than one recipe for a given item, the Oracle Retriever returns possible instantiations of each recipe. For example Figure 6, contains an example

where the Oracle Retriever returns three possible recipes for crafting an iron ingot. All recipes are returned in plain text.

The retriever is not conditioned on the current inventory of the agent, as a result it returns all possible recipes for a given item, regardless of whether the agent has the necessary ingredients in its inventory. We can observe this in Figure 6 where recipes 1 and 3 are not feasible given the agent's inventory. This is analogous to querying a knowledge base, where all matching results are returned, regardless of whether they are feasible.

For many recipes, it is possible for multiple valid instantiations to exist. For example, the recipe for crafting planks 🟫 can be done by placing logs in any slots of the crafting grid. As such, placing a log in any slot of the crafting grid is a valid instantiation of the recipe. In such cases, the Oracle Retriever returns a single random instantiation of the recipe.

Ultimately, the Oracle Retriever can be used to simulate a perfect retriever, which allows us to isolate and evaluate how effectively agents can use external knowledge when it's provided without retrieval errors and grounded within the game environment. In real-world applications, RAG systems would need to handle partial matches, ambiguous queries, and potentially irrelevant information. Our implementation provides a best-case scenario baseline against which more realistic RAG approaches can be compared.

### E.2 Minecraft Wiki

While not used in our experiments, we also include a scraped version of the Minecraft Wiki for future work. To collect the Minecraft Wiki, we crawl and process pages containing crafting recipes which are valid in Plancraft. We convert the html and images to a structured Markdown format, removing irrelevant information and formatting the recipes in a consistent manner. We include the Minecraft Wiki in our release to encourage future work on RAG systems that can leverage multi-modal knowledge sources.

## F  Custom R-CNN Model

Figure 13 shows the custom Faster R-CNN model used to predict bounding boxes along with both class and quantity labels from observations of the inventory. The model is trained from randomly sampling Minecraft inventories in Plancraft. Each training image is automatically annotated with bounding boxes around items, along with class and quantity labels. We extend the Pytorch Faster R-CNN resnet50 V2 (Li et al., 2021) model to add an additional classification head for quantity. Our model therefore predicts both class and quantity labels for each bounding box using the same model and underlying representation.

| | Accuracy | | | Slot | | | IoU | | |
|---|---|---|---|---|---|---|---|---|---|
| Threshold | Slot | Type | Quantity | Precision | Recall | F1 | Slot | Type | Quantity |
| 0.25 | 0.99 | 0.92 | 0.90 | 1 | 0.99 | 0.99 | 0.99 | 0.86 | 0.86 |
| 0.40 | 0.97 | 0.91 | 0.90 | 1 | 0.97 | 0.98 | 0.97 | 0.87 | 0.87 |
| 0.50 | 0.95 | 0.90 | 0.89 | 1 | 0.95 | 0.97 | 0.95 | 0.87 | 0.86 |
| 0.75 | 0.86 | 0.85 | 0.83 | 1 | 0.86 | 0.92 | 0.86 | 0.84 | 0.82 |
| 0.90 | 0.77 | 0.76 | 0.75 | 1 | 0.77 | 0.86 | 0.77 | 0.76 | 0.74 |

Table 8: Evaluation results of the custom Faster R-CNN model on initial inventories of the Plancraft dataset.

We evaluate our bounding box model on the initial inventory of the train split of Plancraft in Table 8 to select a confidence threshold. The confidence threshold is used to filter out low-confidence bounding-box predictions from the model. If two bounding boxes map to the same inventory slot, we keep the one with the highest confidence. We report the following metrics: accuracy for slot, type and quantity (exact match); slot precision, slot recall, and slot F1 score; and the Intersection over Union (IoU) for slot, type, and quantity predictions. We select a threshold of 0.25 for all evaluations as it provides the highest F1 score for slot predictions, while maintaining high accuracy in its predictions.

Few-Shot Examples

```
User:  Craft an item of type:  andesite
inventory:
- diorite [I18] quantity 1
- cobblestone [I30] quantity 1
Assistant:  search:  andesite
User:  Recipes to craft andesite:
recipe 1:
diorite at [B1]
cobblestone at [B2]
Assistant:  think:  To solve this task I need to craft andesite
using 1 diorite and 1 cobblestone side by side.
User:  Ok
Assistant:  move:  from [I18] to [B1] with quantity 1
User:  Craft an item of type:  andesite
inventory:
- diorite [B1] quantity 1
- cobblestone [I30] quantity 1
Assistant:  think:  Now I need to move the cobblestone into
position [B2] to be right of the diorite.
User:  Ok
Assistant:  move:  from [I30] to [B2] with quantity 1
User:  Craft an item of type:  andesite
inventory:
- andesite [0] quantity 1
- diorite [B1] quantity 1
- cobblestone [B2] quantity 1
Assistant:  think:  Now I can craft the andesite by moving it
from the craft slot [0] to a free inventory slot.
User:  Ok
Assistant:  move:  from [0] to [I6] with quantity 1
```

```
User:  Craft an item of type:  iron_ingot
inventory:
- iron_ore [I36] quantity 1
- cobblestone [I30] quantity 1
Assistant:  search:  iron_ingot
User:  Recipes to craft iron_ingot:
recipe 1:
iron_block at [C2]
recipe 2:
smelt {iron_ore}
recipe 3:
iron_nugget at [A1]
iron_nugget at [A2]
iron_nugget at [A3]
iron_nugget at [B1]
iron_nugget at [B2]
iron_nugget at [B3]
iron_nugget at [C1]
iron_nugget at [C2]
iron_nugget at [C3]
Assistant:  think:  To craft an iron_ingot, I need to smelt
iron_ore into an empty slot.
User:  Ok
Assistant:  smelt:  from [I36] to [I35] with quantity 1
```

Figure 6: Few-shot (two-shot) examples provided to models during evaluation. Core actions (move, smelt) are shown in regular text, while optional actions are color-coded (think, search) based on which tools are enabled.

---
**Environment Feedback**

**Error Messages for Invalid Actions:**

- **Move**
    - `[Source] and [Target] must be different`
    - `[Source] must be [0] or [A1] to [C3] or [I1] to [I36]`
    - `You cannot move items into [0]`
    - `[Target] must be [A1] to [C3] or [I1] to [I36]`
    - `quantity must be between 1 and 64`
    - `Format Error: <error message>. Correct format: 'move: from [Source] to [Target] with quantity N'`
- **Smelt**
    - `[Source] and [Target] must be different`
    - `[Source] must be [0] or [A1] to [C3] or [I1] to [I36]`
    - `You cannot smelt items into [0]`
    - `[Target] must be [A1] to [C3] or [I1] to [I36]`
    - `quantity must be between 1 and 64`
    - `Format Error: <error message>. Correct format: 'smelt: from [Source] to [Target] with quantity N'`
- **Think**
    - `Format Error: <error message>. Correct format: 'think: <thought message>'`
- **Impossible**
    - `Format Error: <error message>. Correct format: 'impossible: <reason>'`
- **Search**
    - `Could not find a recipe by that name.`
    - `Format Error: <error message>. Correct format: 'search: <recipe name>'`

---

Figure 7: Error messages returned by the environment when an agent performs an invalid action. These messages try to guide the agent to correct its outputs to format its actions correctly.

Llama 8B with `think` action (Successful ✓)

**User:** Craft an item of type: light_gray_glazed_terracotta
inventory:
– smooth_quartz_stairs [I3] quantity 57
– ...
– light_gray_terracotta [I29] quantity 1

**Assistant:** think: To craft light_gray_glazed_terracotta, I need 1
light_gray_terracotta and 1 colored glazed terracotta, but I have
none of these items. However, I have colored glazed terracotta
in the form of other colored terracotta, I could smelt it and then
apply a dye.
**User:** Ok
**Assistant:** think: I can smelt the light_gray_terracotta to get
light gray clay which I can then color with a dye to get the light
gray glazed terracotta.
**User:** Ok
**Assistant:** think: I need to find a dye color that is not present
in the inventory to craft light gray glaze.
**User:** Ok
**Assistant:** think: Since I have a variety of items in the
inventory, I can check if there is a dye color that can be crafted
from any of the items present.
**User:** Ok
**Assistant:** think: I can use the black_bed to craft black dye.
Then I can use that black dye to dye the light gray terracotta and
get the light gray glazed terracotta.
**User:** Ok
**Assistant:** move: from [I22] to [I5] with quantity 1
**User:** Craft an item of type: light_gray_glazed_terracotta
inventory:
– smooth_quartz_stairs [I3] quantity 57
– ...
– light_gray_terracotta [I29] quantity 1

**Assistant:** smelt: from [I29] to [I32] with quantity 1

Figure 8: Example of successful planning with the `think` action. The assistant reasons through the crafting process, identifies in the first think step that the light_gray_terracotta needs to be smelted. However its reasons are incorrect, as light_gray_terracotta is directly smelted into light_gray_glazed_terracotta. In this case, the model also moves an unnecessary item to the crafting grid, but eventually generates the correct action. Note the shortest path here would have been a single smelt action.

---

**Llama 8B with `think` action (Unsuccessful ✗)**

**User:** Craft an item of type: light_gray_glazed_terracotta
inventory:
- smooth_quartz_stairs [I3] quantity 57
- ...
- light_gray_terracotta [I29] quantity 1

**Assistant:** think: To craft light_gray_glazed_terracotta, I need 4 light_gray_terracotta and 1 clay. However, I don't have clay. I also don't have light_gray_terracotta, but I have regular terracotta. I can try to craft light gray terracotta first.
**User:** Ok
**Assistant:** think: To craft light gray terracotta, I need to mix gray concrete powder with water. However, I don't have gray concrete powder, I need to craft it first. I need 4 gray concrete powder and 1 water to make light gray terracotta.
**User:** Ok
**Assistant:** think: To craft gray concrete powder, I need to mix gray dye and concrete powder. I have gray dye, but I need concrete powder. I also need water to make concrete powder.
**User:** Ok
**Assistant:** think: I need to craft concrete powder first. I need 4 gravel and 1 water to make concrete powder. I also need gray dye to make gray concrete powder.
**User:** Ok
**Assistant:** smelt: from [I36] to [I35] with quantity 1 (to make water)
**User:** Craft an item of type: light_gray_glazed_terracotta
inventory:
- smooth_quartz_stairs [I3] quantity 57
- ...
- light_gray_terracotta [I29] quantity 1

**Assistant:** think: I need to craft water first. I have nether_quartz_ore, which can be smelted into nether_quartz. I can then use the nether_quartz and clay to make concrete powder, and then mix gray dye with the concrete powder to get gray concrete powder.
**User:** Ok
... (continues with incorrect actions until max steps reached)

---

Figure 9: Example of unsuccessful planning with the `think` action. The assistant hallucinates non-existent crafting dependencies, in this case the requirement for water and concrete powder, and attempts to solve for them. It also takes valid actions, but since valid actions that are not part of the solution are not penalized, the model either continues to take incorrect actions or eventually reaches the maximum number of steps.

Llama 8B with `search` action (Successful ✓)

```
User:  Craft an item of type:  light_gray_glazed_terracotta
inventory:
- smooth_quartz_stairs [I3] quantity 57
- ...
- light_gray_terracotta [I29] quantity 1

Assistant:  search:  light_gray_glazed_terracotta
User:  Recipes to craft light_gray_glazed_terracotta:
recipe 1:
smelt {'light_gray_terracotta'}
Assistant:  think:  To craft light_gray_glazed_terracotta, I need to
smelt light_gray_terracotta into an empty slot.
User:  Ok
Assistant:  smelt:  from [I29] to [I13] with quantity 1
```

Figure 10: Example of successful planning with the `think` and `search` actions. Combining the search action with the think action, the model immediately identifies the correct way to craft light_gray_glazed_terracotta using smelt.

Llama 8B with `impossible` action (Unsuccessful ✗)

```
User:  Craft an item of type:  light_gray_glazed_terracotta
inventory:
- smooth_quartz_stairs [I3] quantity 57
- ...
- light_gray_terracotta [I29] quantity 1

Assistant:  search:  light_gray_glazed_terracotta
User:  Recipes to craft light_gray_glazed_terracotta:
recipe 1:
smelt {'light_gray_terracotta'}
Assistant:  impossible:  There is no light_gray_terracotta in the
inventory, only a single light_gray_terracotta, but it is not in
slot [0] to be smelted.
```

Figure 11: Example of unsuccessful planning with the `think`, `search`, and `impossible` actions. Even though this is the same example as before, and the model successfully calls `search`, because `impossible` is introduced, the model incorrectly ends the task early.

Llama 8B with `impossible` action (Successful ✓)

```
User:  Craft an item of type:  diorite_wall
inventory:
- oak_sign [I2] quantity 11
- stripped_birch_wood [I3] quantity 11
- ...
- grass [I35] quantity 42
Assistant:  search:  diorite_wall
User:  Recipes to craft diorite_wall:
recipe 1:
diorite at [B1]
diorite at [B2]
diorite at [B3]
diorite at [C1]
diorite at [C2]
diorite at [C3]
Assistant:  think:  To craft a diorite_wall, I need 6 diorites
in the crafting grid.  However, I don't have any diorites in my
inventory.  I had crafted an andesite earlier, but I don't see it
in my current inventory.
User:  Ok
Assistant:  impossible:  I don't have the required diorite blocks
to craft a diorite_wall
```

Figure 12: Example of successful planning with the `impossible` action. The assistant correctly identifies an impossible task after confirming the required ingredient (diorite) is missing from the inventory.

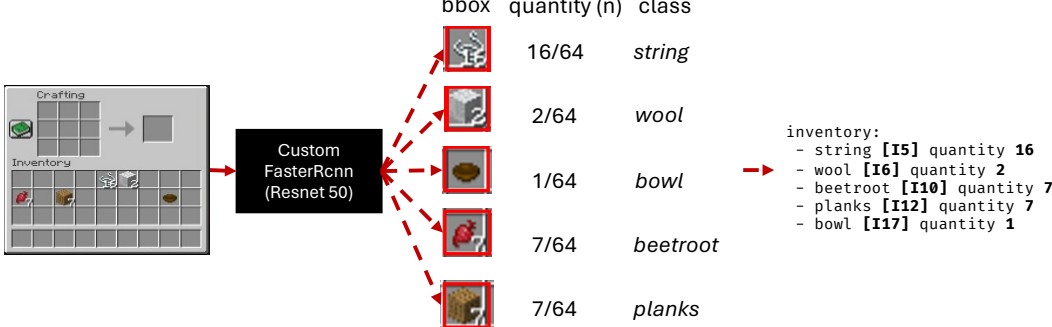

Figure 13: The custom Faster R-CNN model predicts bounding boxes along with both class and quantity labels from observations of the inventory. From the model predictions we can reconstruct a text observation in the same format as provided text-only environment.

