# OpenReview forum: "Plancraft: an evaluation dataset for planning with LLM agents"
_colmweb.org/COLM/2025/Conference — COLM 2025_

### Official Review · Reviewer_ME3a · 2025-05-02

**Rating:** 7
**Confidence:** 4
**Ethics Flag:** 1

**Summary:**

This paper introduces a new planning dataset, Plancraft, designed for LLM agents. Plancraft supports multi-modal inputs based on the Minecraft crafting GUI. It further incorporates an external knowledge base and planner to evaluate whether LLM agents can effectively utilize external tools to solve tasks. Additionally, 17% of the tasks are intentionally unsolvable, aiming to assess the agents’ ability to recognize task infeasibility. Experimental results reveal that both few-shot and fine-tuning inference approaches have their respective limitations, such as noise in visual input interpretation and challenges in generalization.

**Reasons To Accept:**

1.	Plancraft incorporates an external knowledge base and planner to test whether LLM agents can leverage external tools to solve tasks, a setting that is both interesting and important. To some extent, this reflects a more realistic agentic scenario, where the agent is not expected to solve everything independently but instead interacts with well-defined tools. This stands in contrast to prior work, which largely relies on the internal capabilities of LLMs.
2.	The paper is easy to follow, with a well-structured and coherent writing style.
3.	The dataset presented in this paper is of high quality and timely for the community.

**Reasons To Reject:**

1. The "Impossible Tasks" setting is somewhat weird. What exactly is expected from agents when they encounter such tasks? Should they refuse to answer altogether or attempt sub-optimal solutions by, for example, relaxing some constraints in the queries? This aspect is not well explained in the paper, making the setting feel more like a challenge designed to trick the model rather than a realistic scenario that captures meaningful agentic behavior. The analysis is also rather superficial—merely reporting the rate at which agents identify tasks as impossible is insufficient. A deeper exploration of agent behavior in these situations would be more informative.
2. Regarding the evaluated models, both open-source and closed-source, I recommend that the authors consider incorporating a broader range of model families. Relying solely on GPT-family and LLaMA-family models may introduce bias, especially given that the paper fine-tunes a LLaMA 8B model. Including models such as DeepSeek, Qwen, and Gemini would offer a more comprehensive and balanced evaluation.

---

> ### Author Response · Authors · 2025-06-01
> **Response to Reviewer ME3a**
>
> >What exactly is expected from agents when they encounter such tasks? Should they refuse to answer altogether or attempt sub-optimal solutions by, for example, relaxing some constraints in the queries? This aspect is not well explained in the paper, making the setting feel more like a challenge designed to trick the model rather than a realistic scenario that captures meaningful agentic behavior.
>
> As mentioned (L208), the correct response from an impossible task is for the agent to end the episode (emit an impossible action). This is the desired behavior as it means that the agent has correctly identified that the task is unsolvable and therefore does not need to spend time or compute attempting an impossible task.
>
> However, the novelty of introducing the possibility of impossible tasks and the impossible action, means that agents can now declare all tasks to be impossible. This is why we measure impossible as an F1-score (L277).
>
> We believe this is a realistic scenario as an agent deployed in the real world might encounter tasks which are unsolvable and/or rely on invalid assumptions. Knowing when a task is unsolvable is another side of knowing when something is solvable, and therefore we argue that this is a worthwhile feature to test and evaluate in agentic systems. This is especially relevant when the cost of attempting to solve an impossible task is high.
>
> >Regarding the evaluated models, both open-source and closed-source, I recommend that the authors consider incorporating a broader range of model families.
>
> Thank you for this feedback, we are adding Qwen 3 and Gemma 3 to our evaluation table (see our general comment). We hope that this satisfies the reviewer’s demand for a broader range of model families.

---

> > ### Comment · Reviewer_ME3a · 2025-06-07
> >
> > Thanks for your reply. The additional experiments are sufficient, and I look forward to seeing them in the next revision. Considering the completeness of the current version, I will increase the score accordingly.
> >
> > As for Weakness 1, what I want to express is that if agents deliver a sub-optimal plan, how do you process them?  In my opinion, that's not an incorrect answer, since the query itself cannot be completed; therefore, it seems acceptable if the agents relax some constraints.

---

> > > ### Author Response · Authors · 2025-06-08
> > > **Reply to Reviewer ME3a**
> > >
> > > We sincerely thank the reviewer for their follow-up.
> > >
> > > Indeed, in Plancraft's current design, success is binary: the agent must craft the exact item specified. Within this controlled framework, any other plan is a failure, and correctly identifying an impossible task with the `impossible` action is the desired outcome.
> > >
> > > However, we agree that in real-world scenarios, relaxing constraints to find a "next-best" or partial solution could also be a valuable behavior. We appreciate this feedback, as it helps clarify the scope of our work and inspire future directions.

---

### Official Review · Reviewer_rg18 · 2025-05-07

**Rating:** 6
**Confidence:** 4
**Ethics Flag:** 1

**Summary:**

This paper presents the Plancraft dataset, which consists of planning problems, and evaluates a few LLM agents using this dataset. Overall, it is easy to follow and the evaluation methodology is sound. The task has some novelty, but is somewhat incremental to existing datasets for evaluating LLM agents.

The Plancraft dataset is based on the crafting GUI in Minecraft, and has both a text interface and a multi-modal (image) interface. Although there are already existing datasets based on games including Minecraft, this dataset is novel in its inclusion of a set of impossible tasks, which are created by removing essential materials from the inventory. The authors have implemented an expert planner, which finds the most efficient sequence of actions to complete each task. This is used to evaluate the quality (efficiency) of generated plans, in addition to the conventional task success rate. The dataset also includes the Minecraft Wiki, which can test the agent's ability to use external knowledge via RAG.

The authors have compared LLMs on their success rate on different difficulty levels, their efficiency, and their ability to identify impossible tasks early. The use of the "think" and "search" actions (i.e. internal/external knowledge) improves both success rate and efficiency. Another finding is that fine-tuning a small model (with only 1k examples) can increase its success rate, but severely decreases its ability to use new actions. Using image inputs, however, results in severe performance degradation.

**Questions To Authors:**

- Should "0.86% accuracy" on line 262 be 86%?

**Reasons To Accept:**

- Measuring the efficiency (by comparison to the expert planner) and the ability to stop early in impossible tasks is a useful supplement to measuring success rate. This allows more fine-grained comparison between LLM agents.
- There are several useful findings from the experiments. These learnings can potentially transfer to similar tasks and models in future research.

**Reasons To Reject:**

- The contributions are mainly a combination of existing features, except for the impossible set, which is novel.
- The definition of Action Efficiency (AE) is not clear to me. It is described as a ratio but in Equation 1 it seems to be an absolute difference. In its current form, it is less intuitive than just reporting the average plan length of the agent and the expert planner.
- The experiments on image inputs being largely unsuccessful seems to suggest that the task is best modeled using text inputs. The task is unsuitable for evaluating VLMs since it doesn't require visual reasoning and the external knowledge is in text.
- To identify areas for improvements, it would have been useful to analyze why the best model still struggles with the hard examples.

---

> ### Author Response · Authors · 2025-06-01
> **Response to Reviewer rg18**
>
> >The definition of Action Efficiency (AE) is not clear to me. It is described as a ratio but in Equation 1 it seems to be an absolute difference.
>
> To clarify the Action Efficiency (AE) metric, we calculate it by comparing the plan length of the agent to the expert planner only for successfully completed tasks. This is because the agent often fails on hard tasks and long plans, and thus naively comparing the average plan length would make it seem as if an agent is much more efficient, when in fact it is just solving short easy plans when compared to the expert planner. We will add a sentence to clarify this in our main text.
>
> >The experiments on image inputs being largely unsuccessful seems to suggest that the task is best modeled using text inputs. The task is unsuitable for evaluating VLMs since it doesn't require visual reasoning and the external knowledge is in text.
>
> Please see our general response on the failures of VLMs for interactive multimodal planning tasks. We believe that because VLMs cannot solve our task there is still a large gap between the generalisation of reasoning and planning capabilities from pixels in VLMs. We hope that our dataset motivates improvements in VLM planning.
>
> >Should "0.86% accuracy" on line 262 be 86%?
>
> Thank you, that is now corrected

---

> > ### Comment · Reviewer_rg18 · 2025-06-09
> >
> > Thank you for your response. I understand the motivation to exclude failed tasks in Action Efficiency (AE), but the equation doesn't seem to match the textual description of the calculation. I appreciate the insights you shared on the failure of VLMs -- the task does reveal limitations of existing VLMs, e.g. on long sequences and few-shot examples, although formulating the task as a visual one can look a bit artificial when it can be solved in the text modality.

---

> ### Author Response · Authors · 2025-06-10
> **Response to Reviewer rg18**
>
> > I understand the motivation to exclude failed tasks in Action Efficiency (AE), but the equation doesn't seem to match the textual description of the calculation
>
> Ah yes, apologies, we now see the confusion is caused by the phrase "the ratio of the number of actions". We will replace this sentence with "the average difference between the number of actions". Thank you for your feedback.

---

### Official Review · Reviewer_87j3 · 2025-05-11

**Rating:** 7
**Confidence:** 4
**Ethics Flag:** 1

**Summary:**

This paper introduces Plancraft, a multi-modal Minecraft-based dataset for evaluating LLM agent planning. Key features include text/multi-modal interfaces, RAG evaluation via a Minecraft Wiki, a handcrafted planner benchmark, and an "Impossible Set" to test feasibility assessment.

**Reasons To Accept:**

1.Moves beyond success rate to evaluate efficiency, RAG, and the crucial ability to identify unsolvable tasks.
2.Provides useful benchmarks and insights into current LLM planning limitations.

**Reasons To Reject:**

1.While the multi-modal aspect of Plancraft is a key feature, the evaluation of image inputs primarily relies on an R-CNN pipeline for text-based LLMs. I think the paper should include direct evaluations of a broader range of Vision-Language Models (VLMs) (e.g., LLaVA, Qwen-VL) to more fully explore their planning performance on the multi-modal interface.
2.The results for multi-modal inputs, particularly for gpt-4o-mini IMG, show extremely poor performance. While this is an important finding, the analysis of why current VLMs struggle so much in this interactive planning setting feels somewhat cursory. Is it due to the visual grounding, the sequential decision-making, the complexity of the GUI, or a combination? A deeper dive or more targeted experiments could strengthen this aspect.
3.While identifying LLM struggles, the paper offers limited in-depth discussion on concrete methods or future research directions to significantly enhance their core planning abilities beyond general suggestions.

---

> ### Author Response · Authors · 2025-06-01
> **Response to Reviewer 87j3**
>
> We thank the reviewer for their feedback.
>
> >  I think the paper should include direct evaluations of a broader range of Vision-Language Models (VLMs) (e.g., LLaVA, Qwen-VL) to more fully explore their planning performance on the multi-modal interface.
>
> Please see our general response on the failures of VLMs for interactive multimodal planning tasks.
>
> >  Is it due to the visual grounding, the sequential decision-making, the complexity of the GUI, or a combination?
>
> The GUI is simple for humans, but difficult for VLMs as it contains 45 different locations to disambiguate and almost 1k different item types. Plancraft is also out-of-distribution for standard VLMs, which are traditionally trained with photographs rather than game GUIs. As such VLMs fail to ground pixels to specific items and locations.
>
> As we show through our Faster R-CNN experiments, even with a fine-tuned bounding-box model, the overall accuracy falls as the LLM is unable to deal with the small amounts of noise introduced in the location and item types.
>
> >While identifying LLM struggles, the paper offers limited in-depth discussion on concrete methods or future research directions to significantly enhance their core planning abilities beyond general suggestions.
>
> There are many future research directions available to improve the planning of LLMs, least of all, tool use. We hope that the benefits of Plancraft: the impossible set, the multi-modality, and the ability to use external knowledge (through the web/minecraft wiki or the gold recipe search) offers many different angles from which to improve planning capabilities of LLMs and VLMs.
>
> Another benefit of our dataset is that it is dependency free, and while it reproduces exactly the Minecraft GUI, it does so without requiring the game engine and therefore can be distributed as a lightweight package.

---

> > ### Comment · Reviewer_87j3 · 2025-06-09
> >
> > I am generally satisfied with the author's response, and I have modified the score accordingly.

---

### Official Review · Reviewer_Q4bq · 2025-05-17

**Rating:** 6
**Confidence:** 4
**Ethics Flag:** 1

**Summary:**

The paper introduces Plancraft, a multi-modal benchmark for evaluating the planning capabilities of LLM agents using the crafting GUI from Minecraft. The dataset includes both solvable and unsolvable tasks, leveraging the Minecraft Wiki as a knowledge base for Retrieval-Augmented Generation (RAG) and integrating a handcrafted planner for reference trajectories. Tasks range from simple to complex crafting goals requiring symbolic spatial manipulation via actions like move and smelt. The benchmark allows both text and image-based observations, and tests various models (e.g., LLaMA 8B/70B, GPT-4o-mini) under different configurations, including tool use (e.g., think, search) and a dedicated impossible action. Findings show that larger models and tool use significantly improve performance, while smaller models benefit from fine-tuning but fail to generalize to new tools. Importantly, Plancraft is positioned as a controlled, realistic environment to stress test an agent’s planning and feasibility estimation skills beyond simple task success rates.

**Reasons To Accept:**

Plancraft offers a clear and controlled benchmark for evaluating symbolic planning with LLMs, using a low-level action space grounded in a familiar environment.

It includes both solvable and unsolvable tasks, allowing for the evaluation of feasibility judgments—something often missing in prior datasets.

The benchmark supports both text and visual interfaces and includes a handcrafted planner and external knowledge source, enabling rigorous, reproducible comparisons.

**Reasons To Reject:**

The environment is too constrained. It only covers crafting via the inventory interface, and that’s just a small slice of what Minecraft (or planning in general) involves. There’s no exploration, tool usage beyond the basic grid, or real interaction with the broader world. So while the benchmark is clean, it may not reflect the kind of open-ended planning challenges agents face in richer environments like Voyager[1] or UI-TARS[2].

The "impossible" tasks feel a bit underwhelming. They’re made unsolvable by just removing an item from the inventory, which is easy to detect. There’s no need for the agent to plan deeply or reason through a long sequence to realize something can’t be done — it just needs to notice something is missing. That makes the evaluation of feasibility kind of shallow, especially compared to prior work where feasibility depends on constraints like budget or reachability[3].

Minor: Model diversity is lacking. Only two LLaMA models and GPT-4o-mini are evaluated, which doesn’t give a full picture of how other strong VLMs might behave. Also, some findings like “70B is better than 8B” are pretty expected. It would’ve been more interesting if they had dug into why models fail, or whether different interfaces (text vs. image) lead to different strategies or mistakes for the same model.

Minor: Missing related work[3]

[1] Wang et al., Voyager: An open-ended embodied agent with large language models. TMLR 2024.

[2] Qin et al., UI-TARS: Pioneering Automated GUI Interaction with Native Agents. 2025.

[3]: Xie et al., TravelPlanner: A Benchmark for Real-World Planning with Language Agents. ICML 2024.

---

> ### Author Response · Authors · 2025-06-01
> **Response to Reviewer Q4bq**
>
> We thank the reviewer for their feedback.
>
> >The environment is too constrained. It only covers crafting via the inventory interface, and that’s just a small slice of what Minecraft (or planning in general) involves. There’s no exploration, tool usage beyond the basic grid, or real interaction with the broader world.
>
> We understand that Plancraft can feel like a subset of the full Minecraft game. However Voyager (and most other Minecraft environments) do not model the crafting interactions in the game at all. Instead, Wang et al. supply the agents with functional primitives that bypass the crafting interface entirely. Therefore, we view Plancraft as a way to fill the gap in the Minecraft as-a-benchmark literature, where most previous works focus on the “mine” in Minecraft, dealing with navigation and resource harvesting, we focus on the “craft” which is self-contained and deals with how well an agent can plan around its knowledge of recipes and the constraints of its inventory.
>
> Another benefit of our environment is that Plancraft does not require the Minecraft game engine. We replicate the game logic to simulate a pixel-perfect reproduction that only relies on image transformations. This significantly accelerates the speed at which we can evaluate our environment versus other multi-modal Minecraft environments.
>
> We also **do** have tool usage beyond the basic grid: we implement the `search` action as a tool (L233) to explicitly model interactions with external knowledge. Search allows us to measure the upper bounds that a perfect grounded retriever could attain.
>
> >The "impossible" tasks feel a bit underwhelming. They’re made unsolvable by just removing an item from the inventory, which is easy to detect.
>
> Our results show that correctly predicting “impossible” is far from easy. Please see Table 3 in our general response. This shows that, without search, even the 70B model is not able to  predict whether a task is impossible (0.45 F1).
>
> The reviewer is correct that the resource constraint is the only way that examples are impossible in Plancraft. We believe that this is a good start for measuring whether or not models can predict when a task is impossible. Future work could explore other sources of impossibility (such as additional tool constraints, agent constraints, missing tools, etc).
>
> >Minor: Model diversity is lacking. Only two LLaMA models and GPT-4o-mini are evaluated,
>
> Please see our new results with Gemma 3 and Qwen 3 in the general comment to reviewers.
>
> >Minor: Missing related work[3]
>
> We have now included TravelPlanner in our dataset comparison table.

---

> > ### Comment · Reviewer_Q4bq · 2025-06-09
> >
> > Thanks for the detailed reply, sharing the context, and updating the work. I will adjust my score accordingly.

---

### Author Response · Authors · 2025-06-01
**General response to reviewers**

In response to reviewer feedback, we improve model diversity, expand discussion on VLM underperformance in Plancraft, and further justify the impossible set feature.

**Model Diversity**

We add Gemma 3 12B/27B (multimodal) and Qwen3-30B-A3B (MOE+reasoning) as new baselines, released post-submission, to show current LLM performance. For Qwen 3, max generated tokens per step increased (256 to 2048), and remove < think > traces before environment interaction.

**Table 1 (Text)**
| Tools | Model | Easy | Medium | Hard | Overall | Think | Search | Impossible | Impossible F1 | Avg. Plan Length | AE | Tokens Used |
|-------------|------------------|------|--------|------|---------|--------|--------|-------------|----------------|-------------------|------|--------------|
| **M S** | gemma 12B | 0.16 | 0.03 | 0.00 | 0.07	| - | - | - | - | 28.12 | 0.85 | 79.4k |
| **M S** | gemma 27B | 0.24 | 0.08 | 0.01 | 0.12	| - | - | - | - | 27.13 | 1.85 | 67.5k |
| **M S** | Qwen3 30B A3B	| 0.49 | 0.21 | 0.04 | 0.27	| - | - | - | - | 24.29 | 3.66 | 93.6k |
| **M S T** | gemma 12B | 0.20 | 0.06 | 0.00 | 0.10	| 17.53 | - | - | - | 27.82 | 2.85 | 57.2k |
| **M S T** | gemma 27B | 0.38 | 0.20 | 0.00 | 0.20	| 15.56 | - | - | - | 25.70 | 3.56 | 56.4k |
| **M S T** | Qwen3 30B A3B	| 0.34 | 0.23 | 0.03 | 0.20	| 15.49 | - | - | - | 25.61 | 2.90 | 73.1k |
| **M S T SE**| gemma 12B | 0.53 | 0.30 | 0.03 | 0.29	| 13.09 | 1.85 | - | - | 23.75 | 3.36 | 49.8k |
| **M S T SE**| gemma 27B | 0.83 | 0.69 | 0.08 | 0.52	| 8.77 | 2.75 | - | - | 19.57 | 4.06 | 43.6k |
| **M S T SE**| Qwen3 30B A3B	| 0.48 | 0.38 | 0.07 | 0.30	| 11.59 | 0.79 | - | - | 23.60 | 2.84 | 72.9k |
| **M S T SE I**| gemma 12B | 0.43 | 0.28 | 0.02 | 0.25	| 2.91 | 0.58 | 0.71 | 0.36 | 9.05 | 2.87 | 19.1k |
| **M S T SE I**| gemma 27B | 0.78 | 0.62 | 0.07 | 0.48	| 2.73 | 1.22 | 0.51 | 0.45 | 8.90 | 3.80 | 17.5k |
| **M S T SE I**| Qwen3 30B A3B	| 0.32 | 0.28 | 0.03 | 0.20	| 1.69 | 0.15 | 0.68 | 0.38 | 7.88 | 2.33 | 25.5k |

**Table 2 (IMG) (S M)**
| Model | Overall (Accuracy) | Avg. Tokens Used |
|------------------------|--------------------|------------------|
| gemma 3 12B IMG | 0.00 | 137.7k |
| gemma 3 27B IMG | 0.00 | 117.6k |
| Qwen 2.5 VL 72B IMG | 0.01 | 179.7k |
| gpt-4o-mini IMG | 0.02 | 11.6M |

**VLMs in Plancraft**

To solve Plancraft, a VLM would need to discern where and what objects are in an image, how these change over time, and how to map between these and the text locations/types. This is something that all VLMs are currently unable to do in game environments.

As Plancraft can contain dialogue sequences of up to 30 observations, VLMs need to be able to handle 30 different image inputs. When tested in this setting, off-the-shelf VLMs, such as Qwen 2.5 VL, obtain close to 0% task accuracy. This is in part due to the pre-training and fine tuning regime of VLMs which are rarely trained on long image sequences nor are they trained to ground arbitrary game UIs.

Plancraft's complexity also exceeds what current image embeddings capture. There are 45 inventory slots, 979 item types, and up to 64 units per item. VLMs like Gemma 3 26B might approximate positions (e.g., "bottom left") but fail to map items to exact grid locations.

It’s also easy to see that if our own custom Faster R-CNN model, specifically trained on our Minecraft inventories to ground images to the text game state, still entirely fails when used in conjunction with smaller models (Llama 7B), then an off-the-shelf VLM has very little chance to succeed.

Our results also demonstrate that prompting VLMs using few-shot prompting in the same way as with text does not work. The models are unable to leverage the few-shot image examples to calibrate their predictions and generalise more broadly to new tasks. Our hope is that Plancraft pushes research in new directions to either adapt existing VLMs for specific downstream tasks or domains, or to rethink new approaches of leveraging off-the-shelf VLMs for multimodal interactive environments.

**Impossible Set**

We realise that we did not include the results of the models with the impossible set but without search (M S T I).

**Table 3 (M S T I)**
| Tools | Model | Easy | Medium | Hard | Overall | Think | Search | Impossible | Impossible F1 | Avg. Plan Length | AE | Tokens Used |
|-----------|------------------|--------|--------|-------|---------|--------|--------|-------------|----------------|-------------------|------|--------------|
| **M S T I** | Llama 70B | _0.45_ | _0.29_ | _0.03_ | 0.26	| 6.98 | - | 0.43 | 0.45 | 16.81 | 2.75 | 40.3k |
| **M S T I** | gemma 27B | 0.30 | 0.15 | 0.00 | 0.16	| 2.82 | - | 0.82 | 0.34 | 7.68 | 2.89 | 14.0k |
| **M S T I** | Qwen3 30B A3B	| 0.29 | 0.17 | 0.01 | 0.16	| 3.04 | - | 0.71 | 0.37 | 8.72 | 1.88 | 25.1k |

These confirm that predicting whether or not something is impossible is not easy, the best performing model (Llama 70B) only obtains 0.45 F1.

---

### Decision · Program_Chairs · 2025-07-08

**Decision:**

Accept

**Comment:**

This paper introduces Plancraft, a Minecraft-based dataset for evaluating LLM agent planning. There are already some Minecraft-based agent datasets, but Plancraft differentiates by focusing on the 'craft' process in Minecraft while existing efforts usually abstract that away. Compared with non-Minecraft-based planning datasets, Plancraft features a search space with a very large branching factor and out-of-distribution knowledge (w.r.t. typical LLM training) that may necessitate RAG (from Minecraft Wiki as a knowledge base). It also includes an interesting set of 'impossible' tasks to test whether an agent can correctly detect such cases. Evaluation shows that current (multimodal) LLMs generally struggle under this new planning setting.

I think this is a solid contribution even among the quickly growing body of work on LLM/agent planning evaluation. The carefully chosen setting embodies a unique combination of features (e.g., OOD knowledge, large branching factor, impossibility detection) that renders a valuable resource for agent planning research. Most of the concerns raised by the reviewers are addressed during rebuttal.